Effect of circadian rhythm disruption on benign prostatic hyperplasia in rats

Huang Xiaoxue 1
Tang Xiaohu 2
Xu Yuanzhao 2 3
Liu Zhiyan 4
Luo Guangheng luoguangheng1975@126.com 2
1 Guizhou University, GuiZhou University Medical College , Guiyang , Guizhou , China
2 Department of Urology Surgery, Guizhou Provincial People’s Hospital , Guiyang , Guizhou , China
3 Zunyi Medical University, School of Clinical Medicine , Zunyi , Guizhou , China
4 Guizhou Medical University, School of Clinical Medicine , Guiyang , Guizhou , China
Anson Lesley
Electronic publication date: 2025 Oct 15
Publication date: 2025
Volume: 13
Electronic Location ID: e20173
Received 2025 Mar 31; Accepted 2025 Sep 11
Copyright: ©2025 Huang et al.
Copyright year: 2025
Copyright holder: Huang et al.
License: This is an open access article distributed under the terms of the Creative Commons Attribution License, which permits unrestricted use, distribution, reproduction and adaptation in any medium and for any purpose provided that it is properly attributed. For attribution, the original author(s), title, publication source (PeerJ) and either DOI or URL of the article must be cited.
License URL: https://creativecommons.org/licenses/by/4.0/

Keywords: Benign prostatic hyperplasia, Circadian rhythm, Estrogen, Bioinformatics, Light, Testosterone

Funding: National Natural Science Foundation of China 82260157 This work was supported by the National Natural Science Foundation of China (No. 82260157). The funders had no role in study design, data collection and analysis, decision to publish, or preparation of the manuscript.

==============================
Benign prostatic hyperplasia (BPH) is a common condition in middle-aged and elderly men. Disrupted circadian rhythms (CRD) can directly influence aging, inflammation, metabolic syndrome, and hormonal changes—all of which are closely linked to BPH. This study aimed to investigate whether CRD accelerates prostatic hyperplasia in rats. Twenty male Sprague-Dawley (SD) rats were divided into two batches. A BPH model was established using mixed slow-release pellets of testosterone (T) and estradiol (E2). CRD was induced by continuous light exposure (Cle), while a 12-hour light/12-hour dark cycle defined the control (Con) group.

First batch

Rats were divided into T+E2 and T+E2+Cle groups. Initial and final body weight, prostate weight, and prostate index (PI) were recorded. Hematoxylin and eosin (H&E) staining was performed. Serum levels of dihydrotestosterone (DHT) and estradiol (E2) were measured by ELISA, and mRNA expression of circadian rhythm genes was assessed via qRT-PCR.

Second batch

Rats were divided into Con and Cle groups. Body weight, prostate weight, and PI were recorded. H&E staining was used for pathological analysis. Ki-67 expression was assessed by immunohistochemistry (IHC). RNA sequencing (RNA-Seq) was used to investigate gene expression in prostate tissue, validated by qRT-PCR. Differentially expressed genes (DEGs) were analyzed using bioinformatics methods.

First batch results

CRD significantly increased prostate weight, PI, and epithelial thickness; elevated serum DHT levels; and reduced E2 levels. qRT-PCR confirmed that CRD altered circadian gene expression.

Second batch results

CRD significantly increased PI and Ki-67 expression in the prostate. GO analysis revealed significant enrichment in immune response, external side of plasma membrane, and carbohydrate binding (p < 0.001). Kyoto Encyclopedia of Genes and Genomes (KEGG) pathway analysis showed enrichment in cytokine-cytokine receptor interaction, viral protein interaction with cytokine and receptor, phenylalanine metabolism, and chemokine signaling pathways (p < 0.001). Gene set enrichment analysis (GSEA) indicated positive enrichment in voltage-gated calcium channel activity and type II diabetes mellitus. Protein–protein interaction (PPI) network analysis identified Itgad, Ccr7, CD27, Sell, CD69, Gzmb, IRF8, and KIrd1 as highly correlated genes.

Conclusion

These findings suggest that CRD may accelerate prostate cell growth by modulating immune and inflammatory responses, contributing to the development of benign prostatic hyperplasia.

Introduction

Benign prostatic hyperplasia (BPH) is a common urological disease in middle-aged and older men (Gharbieh, Reeves & Challacombe, 2023), mainly due to the proliferation of prostatic epithelial cells and stromal cells (Tamalunas et al., 2021). It can lead to bladder outlet obstruction, which in turn causes a range of lower urinary tract symptoms, such as frequent urination, urgency, difficulty urinating, etc., (Woo et al., 2022). The prevalence of the disease increases with age, and the number of people suffering from BPH is on the rise worldwide, with some studies suggesting that 30–40% of men will develop BPH by age 40, while the prevalence can increase to 70–80% in men over 80 years of age (Song et al., 2023a; Song et al., 2023b). While bringing inconvenience to the lives of elderly men, it also increases the economic burden of society. Internationally recognized pathogenesis includes: hormones, including androgens and estrogens, growth factors, inflammation, autophagy, and oxidative stress. According to a recent meta-analysis, the lifetime prevalence of BPH is estimated to be 26.2%, regardless of ethnic background (Mu et al., 2023). A correlation between BPH and circadian rhythm disruption (CRD) has been reported in the literature. Since the two have common pathogenesis factors, circadian rhythm disorders can lead to inflammation, metabolic disorders, and hormonal disorders, and also affect the occurrence of BPH (Cavanaugh et al., 2024).

Circadian rhythm refers to the physiology, metabolism, and behavior of the human body in a 24-hour cycle (Shen et al., 2023a; Shen et al., 2023b). The normal functioning of mammalian biorhythms relies on the synergy between the autonomic drive of pacing neurons in the hypothalamic suprachiasmatic nucleus (SCN) and the molecular clock network (peripheral clock) in peripheral tissue cells (Ansu & Knutson, 2023). Mammalian behavior and physiology change rhythms along with circadian rhythms, which include not only sleep-wake cycles, but also eating-fasting cycles, as well as changes in reproductive, neurological, metabolic, endocrine, cardiovascular, and immune functions (Allada & Bass, 2021). Due to the development of the economy and service culture, shift work is becoming more common in today’s society (e.g., healthcare and emergency services, hospitality, transportation, and manufacturing) (Boivin, Boudreau & Kosmadopoulos, 2022). Delayed sleep and excessive light exposure at night can disrupt circadian physiology, resulting in severe circadian rhythm disruptions.

Circadian rhythm disturbances can lead to a variety of problems, such as sleep disturbances (Meyer et al., 2022), metabolic disorders (Schrader et al., 2024) and weakened immunity (Shen et al., 2024). In severe cases, it may even increase the risk of developing a variety of diseases such as mental illness (Nassan & Videnovic, 2022), obesity (Chaput et al., 2023), diabete (Speksnijder et al., 2024), tumors (Fishbein, Knutson & Zee, 2021) and cardiovascular diseases (Ansu & Knutson, 2023), etc. Acute circadian disruption caused by constant light exposure promotes the activation of caspase 1 in the hippocampus in mice (Ketelauri et al., 2023). Activation of caspase-1 plays a key role in inflammation (Sharma & Kanneganti, 2021), cell proliferation (Jing et al., 2023), oxidative stress (Zhou et al., 2024), and more. A Mendelian randomized study of the causal relationship between lifestyle habits and benign prostatic hyperplasia showed a strong correlation between sleep levels and BPH, with adequate sleep time being a protective factor against BPH (Jia et al., 2024a; Jia et al., 2024b). Research has shown that disrupted circadian rhythms can affect the function of immune cells. There is a significant association between immune cells and BPH (Li et al., 2024). Publicly available genetic data examined the causal relationship between 731 immunophenotypes and BPH risk. It was found that 38 immunophenotypes had a causal effect on BPH. Studies have shown that the circadian gene Bmal1 regulates prostate growth by regulating the cell cycle (Ueda et al., 2022). Therefore, these findings together suggest that photon interference-induced circadian rhythm disorders may play an important role in the progression of BPH. However, the effects and underlying mechanisms linking circadian rhythm disruption to BPH progression remain largely unknown so far.

In this paper, BPH rat model was established using mixed slow-release pellets of testosterone (T) and estradiol (E2). Circadian rhythm disruption in rats was induced by continuous light exposure (Cle). Two batches of four different groups of animal experiments were conducted. We first conduct experiments on two groups of animals: T+E2 group and T+E2+Cle group. With the aim of investigating whether disruption of the circadian rhythm accelerated the progression of BPH. Subsequently, we divided the animals into two additional groups to carry out the second batch of animal experiments: Con group and Cle group. 12 h illumination: 12 h darkness was defined as a Con group. The purpose is to investigate whether circadian rhythm disturbance has an impact on the growth of the prostate. This article aims to explore whether circadian rhythm disruption affects the progression of BPH.

Materials & Methods

Animal

Twenty adult male Sprague-Dawley (SD) rats, weighing of 350 g ± 10 g, were taken from Chongqing TengXing Biotechnology Co. Ltd, China. Animals were sheltered under control environmental conditions, constant temperature (24 ± 2 °C), humidity (50%∼70%). Animals have free access to standard laboratory feed and sterile water. All procedures involving animals were approved by the Ethics Committee of Guizhou Medical University, Guizhou Province (Ethics Approval Number: 2305082).

Experiment design

After 1 week of acclimatization, rats were randomly allocated into four groups (n = 5) as follows:

First batch:

1. mixed slow-release pellets of testosterone (T) and estradiol (E2) group:T+E2 group;

2. mixed slow-release pellets of testosterone (T) and estradiol (E2) and continuous light exposure group:T+E2+Cle group.

Second batch:

1. 12 h illumination: 12 h darkness group: Con group;

2. continuous light exposure group: Cle group.

After preparing the dorsal skin of the rats, the rats in the T+E2 group and T+E2+Cle group received a subcutaneous local anesthetic with 1% lidocaine and a pill was placed at the incision. The pellet in each rat was replaced with 1 new slow-release pellet each month. With the help of a pellet press, T+E2 is made into a cylindrical solid pill with a diameter of about two mm and a length of about three mm, with a total weight of about 28.6 mg (T: E2 = 10:1). At the same time, the T+E2+Cle group received 24 h of continuous light group, while the T+E2 group placed only pills. The Con group experienced a 12-hour light and 12-hour dark cycle, while in the Cle group, 24 h of uninterrupted lighting were performed. Referring to the literature (Eum et al., 2023) on the circadian rhythm disorder modeling method in mice, rats were given uninterrupted light for 24 h for three months. After 12 weeks, three rats in each group were selected for observation data statistics and prostate histopathological examination. First batch: The initial and final body weight, prostate weight and prostate index (PI) values of rats were recorded. Serum from the T+E2 and T+E2+Cle groups of rats to determine the levels of dihydrotestosterone (DHT) and estradiol (E2). Prostate tissue was stained with hematoxylin and eosin (H&E). mRNA expression levels of core circadian rhythm genes were detected by real-time quantitative reverse transcription polymerase chain reaction (qRT-PCR) (n = 3). Second batch: The initial and final body weight, prostate weight and prostate index (PI) values of rats were recorded. Prostate tissue was analyzed by histopathological staining with H&E, and immunohistochemistry was used to detect Ki67 protein. RNA sequencing (RNA-Seq) was performed on the ventral lobe of the prostate, and qRT-PCR was performed to verify the accuracy of the sequencing results (n = 3). The procedure as presented in Fig. 1.

Figure 1 Summary of the experimental procedure.

Collection of prostate specimens

12 weeks later, the rats were weighed for the last time, and they were euthanized by an overdose of amobarbital (150 mg/kg, intraperitoneal injection). Death was confirmed by the absence of a heartbeat and corneal reflex. The prostate tissues of the rats were immediately removed. The prostate is weighed and the ratio of prostate weight to body weight is calculated. Tissue pieces of the ventral lobe of the prostate were removed from the four groups and stored in RNA preservation solution for qRT-PCR detection. Among them, the Con group and the Cle group respectively removed the ventral prostate (VP) lobes for subsequent transcriptome sequencing. Immediately after these steps, four sets of prostate tissue are placed in 10% neutral formalin.

Serum DHT and E2 were assessed by ELISA

Serum was collected from rats in the T+E2 and T+E2+Cle groups after 12 weeks of slow-release pellets of intervention. Blood samples from rats were collected and centrifuged at 3,000 × rpm for 10 min. DHT (E-EL-0031; Elabscience) and E2 (E-OSEL-R0001; Elabscience) were then detected using an ELISA kit according to the manufacturer’s instructions.

Hematoxylin and Eosin (H&E)

After 48 h of formalin immersion, the prostate tissue underwent conventional paraffin embedding after gradient alcohol dehydration of 70% ethanol, 85% ethanol, 95% ethanol, 100% ethanol. Four sets of paraffin-embedded fixed tissue sections were cut to five µm thickness, deparaffinized and rehydrated, and H&E stained (Beyotime) according to the manufacturer’s instructions. After embedding in neutral resin, observe under an optical microscope and perform measurement and analysis using ImageJ (1.8.0/1.54 g) software. In each selected field of view, select three different acini for measurement, using the software’s ruler tool to measure the distance from the prostate acinar basement membrane to the top of the epithelial cells, which is the epithelial thickness. Record the epithelial thickness value for each acinus and calculate the average for each section.

Immunohistochemistry

For immunohistochemistry, two sets of paraffin-embedded prostate sections (three µm) from Con and Cle were deparaffinized, rehydrated, blocked, and incubated with anti-ki67 antibody (1:500; Boster Biological Technology, Wuhan, China) overnight at 4 °C to assess cell proliferation activity. Sections were then washed with PBS containing 0.1% Triton and incubated with HRP-conjugated secondary antibody (ZSGB Biotechnology, Beijing, China) for 1 h at room temperature. After the last wash with PBS/Triton, sections were stained with DAB (ZSGB Biotechnology, Beijing, China) substrate and hematoxylin. Acquire images of stained sections by brightfield microscopy and quantify the positive stained areas (%) in the images using ImageJ. The same staining without primary antibody was used as a negative control.

RNA-Seq

Total RNA (n = 3 for each group) was extracted from Con and Cle prostate tissues using Trizol Reagent, and RNA quality was assessed by Nano Drop ND-1000 (concentration > 50 ng/µL) and Bioanalyzer 2100 (RIN > 7.0, total RNA > 1 µg). Two rounds of Poly(A) mRNA enrichment were performed using Dynabeads Oligo(dT) (#25-61005, Thermo Fisher), cDNA libraries were constructed using the TruSeq RNA Sample Prep Kit and sequenced on an Illumina HiSeq 2000 System (Illumina) (Shanghai Biotechnology Co.). The statistical power of this experimental design, calculated in RNASeqPower is 0.8. During the experiment, both biological and technical replicates were conducted three times.

Differential gene analysis

Differentially expressed genes (DEGs) were identified using the R package “DESeq2” from the Cle and Con groups (version 3.2.0), where genes that met the q-value < 0.05 and the foldchange > 2 or foldchange < 0.5 thresholds were defined as DEGs. The “ggplot2” package is used to plot correlated volcano maps and heatmaps in R.

Gene ontology and Kyoto Encyclopedia of Genes and Genomes analysis

In order to further explore the biological processes and signaling pathways of these DEGs, functional analyses were performed. Gene ontology (GO) is an integrated bioinformatics initiative for computational analyses on the biological process, cellular component, and molecular function across different species. Kyoto Encyclopedia of Genes and Genomes (KEGG) pathway analysis is used to explore important pathways related to DEGs, which have prognostic significance (Kanehisa et al., 2023). The false discovery rate < 0.05 was considered statistically significant.

Gene set enrichment analysis

Traditional enrichment analyses based on hypergeometric distributions rely on genes that are significantly up-or down-regulated, which often misses genes that are not significantly differentially expressed but are biologically important. GSEA is a computational method that determines whether a set of a priori deviated gene sets shows statistically significant, consistent interference between two biological states (Reimand et al., 2019). Thus, gene set enrichment analysis (GSEA) analysis of the GO and KEGG pathways using clusterProfler’s dataset allows for the examination of gene collections without specifying explicit differential gene thresholds.

Protein–protein interaction network analysis of DEGs and correlation analyses

The Search Tool for the Retrieval of Interacting Gene (STRING) database (https://string-db.org) is an online resource focused on the comprehensive interactions of lists of proteins and genes (Szklarczyk et al., 2023). Cytoscape (version 3.8.0), a free visualization software, has been used for visualizing the protein–protein interaction (PPI) network (Otasek et al., 2019). DEGs were analyzed using STRING, and the parameters for network construction were set as follows: organisms, rattus norvegicus, composite score threshold, 0.4. A PPI network of DEGs was built using the STRING database and subsequently was visualized with Cytoscape.

Considering that too many central genes can lead to a network that is too complex to decipher, too few central genes may miss important information. we used the five algorithms(MCC, MNC, degree, radiality, stress) in Cytoscape’s plugin “Cytohubba” (Chin et al., 2014) to identify the top 30 genes. GeneMANIA (http://genemania.org) is a web site for generating hypotheses about gene function, analyzing gene lists and prioritizing genes for functional assays. Given a query gene list, GeneMANIA finds functionally similar genes using a wealth of genomics and proteomics data. We used GeneMANIA to search for functionally similar genes among the top 30 differential genes obtained from five algorithms.

Quantitative real-time RT-PCR

Total RNA was extracted by Trizol method. The RNA concentration was detected, cDNA was synthesized by reverse transcription according to the instructions of reverse transcription kit, and qRT-PCR was performed using SYBR Green kit (Takara). Quantitative fluorescence analysis was performed using the CFX96 Touch qRT–PCR System (Bio-Rad). The data were analyzed by the 2−ΔΔCt method, and the relative expression levels of genes were expressed as multiples of the relative GAPDH expression levels. All experiments were repeated three times.

Statistical analysis

GraphPad Prism software (Windows version 9.0.0) and ImageJ software were used for data analysis, and the data were normally distributed, and the independent samples T-test was used for data comparison. When P < 0.05, the difference was considered significant (*P < 0.05; **P < 0.01; ***P < 0.001; ****P < 0.0001; ns=no statistical difference, P > 0.05).

Results

Analysis of prostate and serum indexes in rats after circadian rhythm disorder intervention

There was no statistically significant difference (ns) between initial and final body weight in the four groups (Figs. 2B–2C; Figs. S1B–S1C). Prostate weight in the T+E2+Cle group was significantly greater than that in the T+E2 group (P < 0.001) (Fig. S1D), while there was no statistically significant difference in prostate weight between the Cle and Con groups (Fig. 2D). The prostate index (PI) is used to represent the mass of the prostate relative to body weight and is calculated using prostate mass (mg)/body weight (100 g). The prostate index (PI) of T+E2+Cle rats was higher than that of the T+E2 group (P < 0.05) (Fig. S1E). Similarly, the prostate index (PI) of Cle rats was larger than that of the Con group (P < 0.05) (Fig. 2E). However, compared with the T+E2 group, the serum DHT level in the T+E2+Cle group increased significantly (p < 0.05) (Fig. S2A), while the E2 content decreased significantly (p < 0.05) (Fig. S2B).

Figure 2 Analysis of prostate indexes in rats after circadian rhythm disorder intervention.

(A) The morphology of the prostate of rats in each group. (B) Initial body weight of rats in each group. (C) Final body weight of rats in each group. (D) Prostate weights of rats in each group. (E) Prostate index values of rats in each group. *p < 0.05, ns, not significant; when compared with the Con group n = 3.

Histological morphological features

Prostate tissue from each group was collected separately and H&E staining was performed. After continuous light treatment, the epithelial thickness of the prostate in the T+E2+Cle group was significantly higher than that in the T+E2 group (P < 0.05) (Fig. S3). There was no significant difference in prostatic epithelial thickness in the Cle group compared to the Con group (Fig. 3A).

Figure 3 Histological morphological features.

(A) H&E staining for pathological changes of rats’ prostate tissues (left panel), and the prostate thickness of rats (right panel). (B) The distribution and expression of Ki67 in rat prostate tissues (left panel), and the proportion of Ki67 positive cells positive (%) (right panel). Red arrow: prostatic epithelium; Blue arrow: prostatic stroma. Each bar in the graph represents the mean ± S.D. Scale bar = 50 µm, n = 3. *p < 0.05, **p < 0.01, ns, not significant; when compared with the Con group.

Quantitative real-time RT–PCR: Verify the relative mRNA expression of the key 21 genes

In order to illustrate that circadian clock rhythm disorders affect the normal expression of key circadian clock genes in organisms, we selected some circadian genes reported in the previous literature, performed qRT-PCR, and obtained the following results. After light treatment, compared with the T+E2 group, the expression of circadian core clock genes in the T+E2+Cle group was mostly downregulated, the results are in the supplementary documents (Table). Figure S4 shows the mRNA expression levels of these genes.

Immunostaining of the prostate gland of rats with circadian rhythm disorder using Ki-67 antibody

Ki-67 is a nuclear antigen closely associated with cell proliferation, and commonly used as a marker to evaluate tumor proliferation activity. Ki-67 immunostaining was performed on the prostate gland of rats to study proliferative capacity. Counting of Ki67 positive prostate epithelial cells and prostate stromal cells showed that CRD significantly increased the number of epithelial cells and stromal cells in the rat prostate (Fig. 3B).

Differential gene analysis and validation

Transcriptomic analysis revealed significant differences in the expression of 258 mRNAs between the Cle and Con groups, including 70 up-regulated mRNAs and 188 down-regulated mRNAs. Cluster analysis was performed on the two sets of data, and the volcano plot and heatmap of the differentially expressed gene clusters (Figs. 4A and 4B) revealed the differences between the two groups. The development of BPH is not a single “hormone-cell” mechanism, but a polygenic disease driven by “chronic inflammatory-immune imbalance” and “cell cycle control”. In order to explore whether circadian rhythm disorders affect the normal expression of cell cycle and immune and inflammation-related genes, we selected eight related genes, namely Cep55, Espl1, Irf8, Jchain, Kif20a, Mzb1, Pou2af1, and Prc1, and performed qRT-PCR to verify the accuracy of the sequencing results. There was a significant difference between the two groups (P < 0.05), which was the same trend as the sequencing results, indicating that the sequencing results were accurate and reliable (Figs. 4C–4J).

Figure 4 Differential gene analysis and validation.

(A) Volcano plot of DEGs between two comparison groups. Each dot represents one gene: red dots represent the significantly upregulated genes, blue dots represent the significantly downregulated genes, and gray dots represent no significant DEGs. (B) Heatmap of annotated genes. Each column represents a sample, and each gene is visualized in a row red indicates a high abundance, and blue indicates a relatively low abundance of genes. The level of relative mRNA expression: (C) Cep55; (D) Espl1; (E) Irf8; (F) Jchain; (G) Kifa20a; (H) Mzb1; (I) Pou2af1; (J) Prc1; T-tests were used for statistical analysis. *p < 0.05, **p < 0.01, ***p < 0.001, ****p < 0.0001, compared to the Con group.

Functional annotation and pathway enrichment analysis

Further evaluating the functions and mechanisms of these DEGs, For the top 30 upregulated and downregulated DEGs, GO enrichment analysis of biological process terms showed that DEGs were mainly enriched in immune response, defense response to protozoan, and positive regulation of type II. interferon production. GO enrichment analysis of the category cellular component showed that DEGs were mainly enriched in the external side of plasma membrane, cell surface, and endolysosome membrane. GO enrichment analysis of category molecular function showed that DEGs mainly enriched carbohydrate binding, CC chemokine receptor activity and CC chemokine binding (Fig. 5A). The KEGG enrichment analysis of the top 10 upregulated and downregulated DEGs mainly focused on cytokine-cytokine receptor interaction, viral protein interaction with cytokine and cytokine receptor, phenylalanine metabolism, and chemokine signaling pathway (Fig. 5B).

Figure 5 Functional analyses of the differentially expressed genes between Con and Cle group.

(A) GO enrichment analysis for expression of upregulated and downregulated genes. BP, biological process, CC, Cellular component, MF, molecular function. (B) Bubble diagram of KEGG enrichment analysis results of upregulated and downregulated genes.

GSEA analysis

To further illustrate which biological pathways are affected by biorhythm disruptions caused by continuous light exposure, GSEA analyses were performed on genes between the Con and Cle groups. GO enrichment analysis showed that voltage-gated calcium channel activity (GO:0005245) was positively correlated with biorhythm disturbance (Fig. 6A). KEGG enrichment analysis showed a positive correlation between type II diabetes mellitus (rno04930) and circadian rhythm disorders (Fig. 6B). Overall, functional enrichment analysis found that rhythmic disruption was associated with the regulation of signal transduction and metabolism, which was closely related to the development of BPH.

Figure 6 GSEA analysis.

(A) One major GO molecular function gene set after constant light exposure. (B) One major KEGG Pathway changes in Cle rat prostate tissues compared Con group.

PPI network analysis and acquisition of key genes

The PPI network is visualized by Cytoscape, showing 192 nodes and 191 edges (Fig. 7A). Based on the hub gene co-expression network using GeneMANIA, we identified the first three related functions and their interactions with different weights, including 89.53% co-expression, 4.28% colocalization, and 6.19% shared protein domain (Fig. 7B). PPI mapping showed that Itgad, Ccr7, CD27, Sell, CD69, Gzmb, IRF8 and KIrd1 were highly correlated, which could be used as core candidate proteins for validation and function mining.

Figure 7 PPI network analysis and acquisition of key genes.

Screening of circadian rhythm disorders hub genes (A) PPI network of 96 CRD—DEG. (B) The top 30 hub genes and their co-expression genes network.

Discussion

BPH occurs in more than half of older men and is a very common male disease that causes varying degrees of symptoms that interfere with the patient’s quality of life. Early diagnosis and prompt treatment may reduce the progression of BPH. Therefore, there is an urgent need to identify new molecular regulatory mechanisms and therapeutic targets. In recent years, circadian rhythm disorders caused by light interference have become a common global public health problem, and circadian rhythm disorders are mainly phenomena such as shift work and jet lag. This disturbance is closely related to the occurrence and progression of many cardiometabolic, psychiatric, and neurodegenerative diseases. However, the effects of light interference on benign prostatic hyperplasia are still poorly understood.

In the group of BPH rats induced by the imbalanced of estrogen and androgen, the anatomical morphology observed that compared to the T+E2 group, the T+E2+Cle group exhibited an increase in prostate enlargement, prostate weight, and prostate index. The DHT in the T+E2+Cle group was higher than that in the T+E2 group, but the E2 was lower than that in the T+E2 group. Previous studies have shown that with age, DHT gradually decreases while E2 remains unchanged or increases slightly, and an imbalance in the ratio of estrogen to androgen may lead to BPH (Miao et al., 2019). However, this experiment showed that the T+E2+Cle group had more DHT and less E2. Considering that this may be due to a disturbance of hormone metabolism in the body caused by continuous light exposure, it may lead to a decrease in aromatase (CYP19A1) synthesis. Aromatase is a key enzyme in the conversion of testosterone to estrogen, and its expression is regulated by circadian genes such as Clock/Bmal1 (Chu et al., 2019). The study found that knockdown of CLOCK/BMAL1 in peripheral blood mononuclear cells (PBMCs) of patients with polycystic ovary syndrome (PCOS) resulted in a significant reduction in the production of estradiol, meanwhile, by reducing CYP19A1 and upregulating SRD5A1 and SRD5A2 to produce dihydrotestosterone (Johnson et al., 2022). Decreased aromatase (CYP19A1) synthesis may result in a large increase in DHT and a decrease in E2. Circadian rhythm disturbances may trigger a stress response, leading to elevated levels of stress hormones (such as corticosterone), which may inhibit the synthesis of E2. Immune cells and inflammatory factors play an important role in regulating hormone metabolism, and circadian rhythm disturbances may lead to immune system abnormalities, affecting hormone metabolism and secretion. However, the exact cause of this phenomenon is still unknown. Pathological analysis showed that the thickness of acinar epithelial cells in the T+E2+Cle group was higher than that in the T+E2 group, suggesting that circadian rhythm interference may increase the proliferation of prostatic epithelial cells. Circadian clock genes refer to genes that are able to control the biological rhythm of an organism. It regulates not only the mammalian sleep cycle and cognitive function, but also most of its circadian rhythms under physiological conditions (Cai et al., 2023). The qRT-PCR results of the T+E2 group and the T+E2+Cle group showed that there were differences in the expression of circadian genes in rat prostate tissues after the circadian rhythm disorder caused by long-term exposure to light, indicating that circadian rhythm disorder would affect the expression of circadian rhythm genes. The normal expression of core circadian clock genes is closely related to human health. In this pilot trial, we demonstrated that circadian rhythm disruption accelerates prostatic hyperplasia in rats with an imbalanced testosterone/estradiol.

In the rats with circadian rhythm disorder induced by continuous light, anatomical morphology was observed: the prostate index increased in the Cle group compared with the Con group. Pathological analysis showed no significant changes in the thickness of acinar epithelial cells in the Cle and Con group, and the proportion of Ki-67 positive staining in the Cle group was significantly higher than that in the Con group, indicating that the main feature of circadian rhythm disorder rats was prostatic enlargement with epithelial cell hyperplasia. In the long run, this can lead to the occurrence of benign prostatic hyperplasia. This is the first article to focus on circadian rhythm disruptions that may affect the clinical development of benign prostatic hyperplasia trials. In this trial, we demonstrated that circadian rhythm disturbance increased the percentage of Ki-67 cell positivity in rats, indirectly reflecting circadian rhythm disruption accelerating proliferation of prostate epithelial cells. However, we did not observe significant adverse effects of continuous light exposure on rats.

Although the exact pathogenesis of BPH has not been fully elucidated, there is growing evidence that circadian rhythm disturbances influence cell cycle processes, the development of immune and inflammatory responses, and may be involved in the development of BPH. In recent years, the use of molecular biology techniques to find potential biomarkers has become an important means to explore the pathogenesis of most diseases. In this study, high-throughput sequencing was used for the first time to establish a “gene map” of the prostate gland of rats in the continuous light group and normal group, and the differential genes Cep55, Espl1, Prc1, Kif20a, Jchain, Mzb1, Pou2af1, which may be involved in the regulation of cell cycle and immune inflammation were screened out, and qPT-PCR was performed. Cep55, Espl1, Prc1 and Kif20a could regulate the cell cycle and were positively correlated with the proliferation markers PCNA and Ki-67. Cep55 is a potential immunotherapy combination target by regulating cell cycle drive epithelial proliferation while activating the IL-6/JAK/STAT3 inflammatory axis and immunosuppressive microenvironment (Xie et al., 2023). In 18 epithelial-derived tumors, including colorectal cancer, lung cancer, ovarian cancer, and prostate cancer, the levels of Espl1 mRNA and protein were significantly higher than those in adjacent normal tissues, and were positively correlated with Ki-67 index and S/G2 cell ratio (Zhong et al., 2023). Espl1 forms a PPI network with CDK1, CCNB1, PTTG1 and other core cell cycle genes, suggesting that it drives epithelial cell proliferation through the classical cell cycle axis (Nie et al., 2022). Circadian rhythm dysregulation leads to upregulation of Prc1, which drives excessive cell proliferation by affecting the overactivity of the G2/M phase of the cell cycle (Zhao et al., 2024). Kif20a has been shown to promote the malignant proliferation of multiple epithelial cells through cell cycle drive and plays a key role in prostate cancer (Moon, 2024). Clinically, high expression of Kif20a is significantly associated with higher Gleason scores and PSA levels and worse prognosis (Copello & Burnstein, 2022). Although its direct involvement with BPH is currently lacking, based on its mechanism, Kif20a may be involved in the epithelial/matrix hyperproliferation of BPH via the FOXM1/KIF20A axis. Studies have shown that in the environment of hormone-induced chronic inflammation, Jchain (immunoglobulin linkage chain) expression is upregulated in multiple epithelial layers outside the cervix and co-expressed with genes related to cell proliferation, suggesting that Jchain may be involved in inflammation-induced epithelial cell proliferation (Kaldhusdal et al., 2025). Mzb1 (marginal zone B cell protein) is a B-cell-specific and endoplasmic reticulum localization protein that enhances immune cell infiltration (CD4+/CD8+ T cells, NK cells) and inhibits ovarian cancer cell migration and proliferation, and can be used as a potential immunotherapy target and prognostic marker (Zhu et al., 2025). There may be a synergistic effect between Pou2af1 (B cell coactivator) and multifactors, which jointly regulate the activity of immune cells such as T cells, B cells, and NK cells, thereby affecting the tumor immune microenvironment. The findings suggest that Pou2af1 deficiency impairs the immune response and alters gut microbiota composition, exacerbating DSS-induced colitis (Huang et al., 2025).

RNA-seq results showed that a total of 268 DEGs were found. In the analysis of the results of GO enrichment, we found that the down-regulated differentially expressed genes were significantly enriched in the immune and inflammatory response processes, which is consistent with the previous findings that immune inflammation is associated with the development of BPH (Bostanci et al., 2013; Bostanci et al., 2013). Weakened immune function predisposes to chronic inflammation. Chronic inflammation is involved in the development of BPH and is thought to be a pathogenesis of prostatic hyperplasia (Silver et al., 2024). Immune inflammation promotes prostatic hyperplasia by affecting the number of inflammatory cells and immune cells of prostate cells (Song et al., 2023a; Song et al., 2023b), increasing immune response mediators (Kinsel et al., 1989) and activating MAPK signaling pathways (Yang, Xu & Zhuang, 2020). Circadian rhythm disruptions reduce the body’s immune response by affecting a variety of mechanisms such as immune cell function, differentiation, migration, and inflammatory response (Zeng et al., 2024; Jerigova, Zeman & Okuliarova, 2022). Circadian rhythm disruptions can also reduce the number and activity of T-regulatory cells, which inhibit autoimmune responses and inflammation, leading to worsening of autoimmune diseases. Immunotherapy may be a new target for the treatment of BPH, as in a previous Mendelian randomized study, which showed that specific immune cell phenotypes were significantly associated with the risk of benign prostatic hyperplasia. 38 immunophenotypes had causal effects on BPH (Li et al., 2024). The GO term enriched in the up-regulated differentially expressed genes was mainly in cell differentiation. Previous studies have shown that over-activated transforming growth factor-β-1 (TGF-β1) exacerbates benign prostatic hyperplasia by triggering epithelial-mesenchymal transition (EMT) and epithelial and stromal cell differentiation (Kim, Jin & An, 2023). Myofibroblasts in the prostate play a key role in tissue repair and fibrosis. Under normal circumstances, the activation of myofibroblasts is necessary for wound healing, however overactivation of myofibroblasts leads to tissue fibrosis, which is closely related to the development of BPH (Sampson, Berger & Zenzmaier, 2012). These results suggest that circadian rhythm disruption promotes the development of BPH by influencing mechanisms such as cell differentiation, inflammatory response, and cell proliferation.

In addition, the results of KEGG analysis showed that cytokine-cytokine receptor interactions are the most important signaling pathways, which have been shown to have effects on chronic inflammation (Jia et al., 2024a; Jia et al., 2024b), immune processes (Kang et al., 2022), proliferation (Huang et al., 2023) and migration (Bi, Huang & Liu, 2019) in previous studies. The up-regulated differentially expressed genes were mainly concentrated in the antimicrobial peptide-activated protein kinase (AMPK) signaling pathway and estrogen signaling pathway. The AMPK signaling pathway coordinates cell growth, autophagy, and metabolism (Mihaylova & Shaw, 2011). The AMPK signaling pathway has previously been shown to be associated with BPH (Li et al., 2023; Wang et al., 2025; Geng et al., 2016), upregulation of the AMPK signaling pathway can promote the proliferation of prostatic epithelial cells. BPH is a hormone-dependent disorder that castrated individuals do not develop BPH. Androgens and estrogens are involved in the development of BPH, and estrogen signaling also plays an important role in the pathophysiological process of prostatic hyperplasia, and estrogen can regulate the proliferation of primary mesenchymal cells and the expression of inflammatory factors during BPH (Yang et al., 2022). The up-regulated and down-regulated differentially expressed genes were mainly concentrated in the cytokine-cytokine receptor interaction signaling pathway, suggesting that these pathways may play an important role in circadian rhythm disruption affecting prostate growth. To further investigate the genomic differences caused by circadian rhythm disruptions in rats, we performed GSEA enrichment analysis. The GSEA results showed that the enrichment process of GO and KEGG was mainly in voltage-gated calcium channel activity and Type II diabetes mellitus. Type 2 diabetes is a metabolic disease, and working night shifts is associated with an increased risk of developing type 2 diabetes (Vetter et al., 2018; Shan et al., 2018). Studies have shown that it has the same influencing factors as the development of BPH, and that type 2 diabetes may exacerbate the development of BPH (Johnstone et al., 2021; Yang et al., 2024). These results suggest that rhythmic disruption is associated with the regulation of cell proliferation, migration, and adhesion, which is closely related to the development of BPH.

Some genes obtained by transcriptome sequencing were selected through PPI network interaction. PPI protein interaction mapping showed that Itgad, Ccr7, CD27, Sell, CD69, Gzmb, IRF8 and KIrd1 had high correlation degrees, which could be used as core candidate proteins for verification and function mining. The top eight genes were selected and discussed as follows. Ccr7, CD27, Sell and CD69: all T cell markers. Among them, Ccr7 is involved in the homing of T cells to lymph nodes, regulating T cell migration and immune response initiation (Liu et al., 2021). CD27 promotes T-cell survival and proliferation (Starzer & Berghoff, 2020). Sell regulates T-cell migration and tissue infiltration at sites of inflammation (Yuan et al., 2024) . CD69 is involved in T cell activation and initiation of immune responses. CD69 is positively correlated with T cells. A decrease in T cells can lead to an imbalance in the inflammatory response, leading to chronic inflammation or tissue damage (Li et al., 2024). Itgad, also known as Cd11d, mediates cell adhesion and inflammatory signaling, and a significant upregulation of Cd11d expression was detected in pro-inflammatory macrophages, presumably responsible for macrophage accumulation at the site of inflammation and exacerbation of chronic inflammation (Shen et al., 2023a; Shen et al., 2023b). In bladder cancer cells, KIrd1 mainly enriched in extracellular stromal tissues, is positively regulated by cell proliferation (Liu et al., 2023a; Liu et al., 2023b). The study found that Gzmb plays an important role in obesity-related inflammation by influencing adipose tissue inflammation and insulin resistance (Cimini et al., 2020). IRF8 directly regulates the transcription of multiple genes and is an important regulator of macrophage, dendritic cell (DC), and B cell development, and has been implicated in Th17, Th9, and Treg cell differentiation (Anderson et al., 2021; Luo et al., 2022). Genome-wide association studies (GWAS) have shown that sequence variants in the IRF8 gene are important risk factors for a variety of chronic inflammatory diseases in humans (Salem, Salem & Gros, 2020). Genemania predictor genes are also mostly involved in the regulation of immunity, inflammation. For example: ZAP70, too little, too much can lead to autoimmunity (Ashouri et al., 2022). PRF1 is closely linked to the immune microenvironment (Liu et al., 2023a; Liu et al., 2023b). CD40-mediated activation of macrophages and dendritic cells in mouse intrahepatic cholangiocarcinoma significantly enhances the response to anti-PD-1 therapy (Diggs et al., 2021). At present, there is limited direct research on the association between Itgad, Ccr7, CD27, Sell, CD69, Gzmb, KIrd1 and IRF8 genes and BPH. However, they are mostly related to immune and inflammatory regulation and may play a role by regulating the immune microenvironment of BPH.

In this experiment to investigate whether circadian rhythm disorder affects the development of BPH, we first induced BPH in rats using estrogen and androgen, while continuous light disrupted the circadian rhythm of rats. By comparing the anatomical and pathological staining results of BPH rats and BPH combined with circadian rhythm disorder rats, we found that circadian rhythm disorder accelerated the development of BPH. In order to exclude the effects of exogenous estrogen and androgen on prostate growth, we only performed high-throughput sequencing on the prostate of normal and continuous light exposed rats to explore the differentially expressed genes, biological processes, and signaling pathways that affect prostate cell growth due to circadian rhythm disorders. This provides a foundation for future research. These data enable future researchers to access a subset of genes in rats with rhythm disorders. However, our research still has some limitations. Firstly, we did not consider performing Ki-67 immunohistochemical staining on the first batch of hormone drug treated rats to investigate whether circadian rhythm disorder also increased the expression level of Ki-67 protein in estrogen and androgen induced BPH rats. Secondly, we did not perform immunohistochemical staining on immune cells and inflammation related markers to verify the accuracy of high-throughput sequencing results. Finally, in this study, only qRT-PCR was used to validate some differentially expressed genes, and the specific mechanism of the impact of circadian rhythm disorder on prostate growth has not been validated, which requires further research. Therefore, our findings represent a preliminary exploration and further research is needed to fully understand the relationship between temporal changes, rhythm genes, and BPH.

Conclusions

Our study suggests that circadian rhythm disruption may accelerate the growth of prostate cells by modulating immune and inflammatory responses, contributing to the development of benign prostatic hyperplasia.

Supplemental Information

Supplemental Information 1 Analysis of prostate and serum indexes in rats after circadian rhythm disorder intervention

Supplemental Information 2 DHT and E2

Supplemental Information 3 Histological morphological features

Supplemental Information 4 Relative mRNA expression of the key 21 genes

Supplemental Information 5 Differential gene analysis and validation

Supplemental Information 6 Supplemental Figure and Table

Supplemental Information 7 MIQE Checklist

Abbreviations

BPH Benign prostatic hyperplasia

LUTS Lower urinary tract symptoms

T Testosterone

E2 Estradiol

SCN Suprachiasmatic Nucleus

DEGs Differentially expressed genes

Additional Information and Declarations

Competing Interests

Author Contributions

Animal Ethics

Data Availability

The authors declare there are no competing interests.

Xiaoxue Huang conceived and designed the experiments, performed the experiments, analyzed the data, prepared figures and/or tables, authored or reviewed drafts of the article, and approved the final draft.

Xiaohu Tang conceived and designed the experiments, performed the experiments, analyzed the data, authored or reviewed drafts of the article, and approved the final draft.

Yuanzhao Xu performed the experiments, analyzed the data, prepared figures and/or tables, and approved the final draft.

Zhiyan Liu performed the experiments, prepared figures and/or tables, and approved the final draft.

Guangheng Luo conceived and designed the experiments, authored or reviewed drafts of the article, and approved the final draft.

The following information was supplied relating to ethical approvals (i.e., approving body and any reference numbers):

All procedures involving animals were approved by the Ethics Committee of Guizhou Medical University, Guizhou Province (Ethics Approval Number: 2305082).

The following information was supplied regarding data availability:

The raw sequence data are available at NCBI BioProject: PRJNA1230861.

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
