# Peer review of "Effect of circadian rhythm disruption on benign prostatic hyperplasia in rats"

_PeerJ, doi:10.7717/peerj.20173_

## Round 0.1 · original submission · Major Revisions

**Language Note:** The review process has identified that the English language must be improved. PeerJ can provide language editing services - please contact us at [email protected] for pricing (be sure to provide your manuscript number and title). Alternatively, you should make your own arrangements to improve the language quality and provide details in your response letter. – PeerJ Staff

Reviewer 1 ·

Basic reporting

The manuscript submitted by Huang et al. proposed that circadian rhythm disruption may contribute to the pathogenesis of benign prostatic hyperplasia (BPH) in a rat model. The authors performed the animal study by inducing continuous light exposure (Cle) and implanting the steroid pellet (T+E2) to disrupt circadian rhythm. Histopathological examination and immunohistochemistry analysis of Cle group revealed abnormalities compared to the control group. Furthermore, the transcriptomic analysis by RNA-Seq and subsequent bioinformatic analysis, including functional annotations, pathway enrichment, gene set enrichment, and gene network analysis, were conducted to preliminarily investigate the potential gene functions involved in the circadian rhythm disruption contributing to BPH.
The authors addressed an important issue: the cause of circadian rhythm disruption in disease development. However, several points need to be addressed and improved.

Experimental design

Major point:
1. The most confusing and ambiguous aspect of this study is the inconsistency in the experimental groups across different results. For example, the authors conducted the RNA-Seq experiment using the Control and continuous light exposure (CLE) groups (Figure 3); however, they examined the expression of circadian genes using RNA samples from the T+E2 and T+E2+CLE groups (Figure 7).

Validity of the findings

Major points:
2. As mentioned above, similar issues are also found in several other figures:
- Figure 1: What are the DHT and E2 levels in the CON and Cle groups without T+E2 pellet implantation?
- Figure 2: What is the Ki67-positive cell percentage between the T+E2 and T+E2+Cle group?
- Figure 7: What are the relative expressions of key circadian genes of CON and Cle groups without additional T+E2 administration?
3. What is the rationale for using T+E2 to induce BPH?
4. Is GAPDH an appropriate internal control for real-time PCR in circadian rhythm study? According to a previous study (J Biol Chem. 1998; 273:446-52. doi: 10.1074/ jbc.273.1.446.), GAPDH itself is regulated by the circadian rhythm. Therefore, the authors should justify their use of GAPDH as the internal control in this context. Alternatively, they may consider using a more stable reference gene as suggested in other literature, such as BMC Mol Biol. 2010;11:60. doi: 10.1186/1471-2199-11-60.
5. To convincingly demonstrate that continuous light exposure, whether with or without additional hormone administration, disrupted the circadian rhythm in this study. I suggest rearranging Figure 7 as Figure 1. Presenting the circadian gene expression data earlier would help establish the successful disruption of circadian rhythm as a foundational aspect of the experimental model.
6. In the Discussion section, the authors devote a substantial paragraph to how circadian rhythm disruption alters the expression of immune-related genes. However, there is limited discussion on the connection between immune responses and the pathogenesis of BPH. Further elaboration on this relationship would strengthen the interpretation of the findings.
7. Please unify the reference formats in the Reference section. E.g., L453
8. The resolution of figures and the content of figure legends should be primarily improved:
- Figure 1: please increase the resolution of figures and elaborate on the statistical meaning of figures.
- Figure 2: please specify which plane or region of the prostate was used for the paraffin sections shown in panels E–J.
- Figure 3: please revise the legend to reduce redundant wording
- Figure 4: please enhance the resolution and clearly indicate which experimental groups were used for the functional annotation and pathway enrichment analyses.
- Figure 5: please specify the experimental groups used for the GSEA analysis.
- Figure 6: please improve the image resolution and indicate which experimental groups were included in the gene network analysis.

Minor points:
1. The method of RNA-Seq should be described in more detail so that the readers can follow.
2. L 215 “C”ompared should be in lowercase.

Reviewer 2 ·

Basic reporting

This work by Xiaoxue Huang and colleagues examined the gros tissue, and gene expression data between the prostates of rats exposed to continuous light to disrupt their circadian rhythm while mimicking conditions often occurring in human. The hypothesis of this study is that factors leading to circadian rhythm dysregulation would lead to benign prostate hyperplasia.
The study is concise and written in a fear clear manner. It is novel and would be worth publishing but I have several concerns regarding the presentation of the data and the conclusions and I do not think this manuscript can be recommended for publication.

1) The authors should state the reason for supplementing the rats with the T and E2 pads. Why did they not show the experiments in the non-supplemented mice first?
2) The authors collected tissue from 20 prostates but present RNAseq data only from 6 (three tests and 3 controls) and it is not clear what it is compared and validated in figure 3. The Authors should show all the data from all the mice included in the study.
3) The presentation of the data in figure 3 should be reflecting the design of the study and all the generated gene expression data should be presented (not only the 6 shown in figure) and rendered publicly available for the reviewers’ evaluation. For example, the reviewer suggests using volcano plots highlighting adjusted p-values vs FC to better visualize the best candidate genes differentially expressed in each comparison group
4) The authors should be more precise in describing where the samples for RNAseq were taken from the entire prostate. Did the authors select specific anatomical regions of the prostate of the mice or were they randomly selected for further analysis? See line 131
5) Immune cells staining including myeloid, and macrophages Granzymes should be performed as validation of the findings.
6) Fig 2 I, J, K: quantification of Ki67 of staining should show also T and E2 differences, and subsequent slides sections should also show staining immune infiltration with adequate markers for M1, M2 macrophages.

Minor

1) Can the authors explain why the E2 in the serum of the rats decreases? This is counterintuitive because of the use of the patch.

Experimental design

the expression data should be measured from anatomically comparable regions of the prostate. the authors should be more precise in their M&M section

Validity of the findings

the findings should be validated. See point 4,5, 6

·

Basic reporting

- I suggest to standardize the title font and affiliation with the body of the text. Size, font and spacing need to be standardized throughout the article, it is disorganized;

ABSTRACT
- Add the aim of study and relationship with DMT2 as well because the authors made this correlationship in study;
- Include what age, lineage, and years old of animals used;
- Include p value on abstract as well;

INTRODUCTION
- Line 70: Avoid the use of abbreviatons;
- Line 96: The group description is different on abstract;

Experimental design

- I strongly suggest to authors make a timeline as figure to elucidate the study methodology;
- First describe the group name, after use abbreviations;
- Line 97: First describe the group name, after use abbreviations. I know that SD is Sprague-Dawley but the readers maybe not.
- Lines 105-108: A bit confuse here, on abstract they authors said that 4 animals were used, then in last paragraph you described 2 groups, and now 20 SD animals, re-write please. The timeline is necessary to improve the understand;
- What age the animals started the experiment ?
- Lines 116 -117: Wheres the methodology references ?
- Lines 133-134: Is it common? Normally we removed the prostate, 24 OR 36 OR 48 hours of fixation in bouin or formal, followed by alcohol 70% is used to re-hydrates the tissues, anf finally, start the process of inclusion in paraffin or some similar compound;
- Line 134: Can authors cited some reference using this methodology please ?
- Line 136: The authors said: "The rest of prostate", I would like to know which rest? Which specific lobe or gland part ?
- Line 141: Which alcohol percentage were used ?
- Line 143: Harris hematoxylin? I suggest to do a better description here.
- Line 144-145: What methodology were used to mensured the areas ?
- Lines 149-150: The authors should described better this process in last paragraph because is very wide.
- Line 150: Include each antibody, with dilution and brand as well.
- Lines 230-231: Fig I-J are related to KI-67 not H&E, I suggest to improve the explanation here;
- Line 235: KI-67 should be mention at first time.
- Lines 234-236: I suggest the authors to separates the techniques explanation, IHC is not morphological technique is related to protein expression, authors discussed together;
- Lines 242-244: Did the authors mensured the same area of prostate? Because I had a impression that H&E in 2 first figures are more epithelial and 2nd one more in connective tissue labeling.
- How did authors mensured the KI-67 ? What software and plugin ?
- Lines 245-246: Improve this conclusion. The KI-67 was mensured in all prostate area? The 2st picture looks like epithelial labeling but the 2nd one don't represents very well because it looks like connective tissue area, different zones to mensure together.
- Lines 300-301: I recommend joining the 4 groups in the same graph for aesthetic purposes and better understanding of the data. There are many scattered graphs that do not communicate with each other, and the reader loses the line of reasoning.
- Line 321: What specific area ?
- Line 345: Wich prostate area specifically ?
- Line 388: "inflammation, etc", I suggest to remove the "etc" of sentence;
- Lines 406-408: Have no correlation with prostate in this study;
- Line 408: "NAMPT", avoid start a paragraph using abbreviations;

CONCLUSION
- Improve the conclusion, too repetitive.
- Line 443: Messy topic, improve.

FIGURES
- I suggest to authors do a single graph to all analysis, I mean include the 4 groups in only one graph, instead of to compare each both groups separated;
- I would like to see the bars on H&E images;

Validity of the findings

The study has scientific relevance as well as impact in the area of ​​interest, however, modifications need to be made to better elucidate the methodology, the data, and the conclusion found by the authors in question.

Additional comments

Despite the numerous requests for changes to the study, I congratulate the authors for the competent and thorough work they have done so far. On the other hand, I strongly encourage them to review the English (grammar), standardize the fonts, sizes and spacing throughout the entire text, and check the journal's standards so that their study complies. I believe in the replicability of the data, but improvements need to be made before we move forward. In conclusion, reducing the graphs analyzing the 4 groups into a single graph for each analysis makes it easier for the reader and/or reviewer to understand the study performed.

---

## Round 0.2 · Major Revisions

Reviewer 1 ·

Basic reporting

In the revised version, the authors have rearranged the figures in the main text and supplementary information to reduce data inconsistencies. In this study, they conclude that circadian rhythm disruption may accelerate prostate cell growth by modulating immune and inflammatory responses, thereby contributing to BPH development. This conclusion is supported by GO, KEGG, GSEA, and PPI analyses based on transcriptomic results. However, direct evidence linking circadian rhythm, immune function, hormone dysregulation, and BPH remains to be explored in future studies.
Minor points:
1. Please unify the spacing format between words and parentheses throughout the manuscript, including in cited references and abbreviations.
2. Please spell out all the abbreviations when they first appear in the manuscript.
Line 20, Cle or should be labeled in line 19.
Line 23, CON.
Line 40, PPI.
Line 283, CC
Line 382, redundant use of the abbreviation "BPH"
3. Please elaborate on the rationale for selecting these genes as validation (Fig. 4C–J, except irf8), and discuss how their functions may be related to circadian rhythm and BPH in the discussion section.

Experimental design

No comment.

Validity of the findings

No comment.

Additional comments

No comment.

Reviewer 2 ·

Basic reporting

I would like to thank the authors for submitting an improved version of the paper.
the authors have addressed some of my concerns. I suggest however to mention all the shortcomings in the rebuttal in the answers to this reviewer also in the discussion section of the paper.

Experimental design

this has not changes since previous submission

Validity of the findings

same as in the previous submission

·

Basic reporting

I have carefully reviewed the revised version of the manuscript entitled "The effect of circadian rhythm disorder on benign prostatic hyperplasia in rats", as well as the detailed rebuttal letter provided by the authors. I acknowledge and appreciate the effort made by the authors to address the concerns raised during the initial review round.

Experimental design

The revised manuscript demonstrates clear improvements in terms of clarity, methodological transparency, and scientific rigor. The authors have successfully clarified the rationale for the experimental design, enhanced the description of the methods, and expanded on the interpretation of their findings, particularly in relation to the role of circadian rhythm disruption in BPH progression. The additional data and explanations provided help to strengthen the overall narrative and support the conclusions drawn.

Validity of the findings

I have also verified the raw data availability, ethical approval documentation, and image integrity, and found no concerns. The figures and tables have been updated accordingly and meet the journal’s quality standards.

Additional comments

In light of these revisions, I believe the manuscript now meets the criteria for publication. I have no additional suggestions or requests for changes. Therefore, I consider the current version acceptable and recommend its publication.

---

## Round 0.3 · accepted · Accept

Thank you for revising your manuscript to address the reviewers' concerns. Reviewers 1 and 3 have both recommended acceptance and I am also satisfied with the revisions you have made in response to the earlier comments of reviewer 2. The manuscript is now ready for publication.

Reviewer 1 ·

Basic reporting

No comment

Experimental design

No comment

Validity of the findings

No comment